# Stateful Rotor for Continuity of Quaternion and Fast Sensor Fusion Algorithm Using 9-Axis Sensors

**DOI:** 10.3390/s22207989

**Published:** 2022-10-19

**Authors:** Takashi Kusaka, Takayuki Tanaka

**Affiliations:** 1Independent Researcher, Sapporo 063-0867, Japan; 2Graduate School of Information Science and Technology, Hokkaido University, Sapporo 060-0814, Japan

**Keywords:** IMU, MARG, embedded systems, complementary filter, quaternion, state tracking, multiply–add operation, double covering

## Abstract

Advances in micro-electro-mechanical systems technology have led to the emergence of compact attitude measurement sensor products that integrate acceleration, magnetometer, and gyroscope sensors on a single chip, making them important devices in the field of three-dimensional (3D) attitude measurement for unmanned aerial vehicles, smartphones, and other devices. Sensor fusion algorithms for posture measurement have become an indispensable technology in cutting-edge research, such as human posture measurement using wearable sensors, and stabilization problems in robot position and posture measurement. We have also developed wearable sensors and powered suits in our previous research. We needed a technology for the real-time measurement of a 3D human body motion. It is known that quaternions can be used to algebraically handle 3D rotations; however, sensor fusion algorithms for three sensors are presently complex. This is because these algorithms deal with the post-rotation attitude (pure quaternions) rather than rotation information (the rotor) to avoid a double covering problem involving the rotor. If we are dealing with rotation, it may be possible to make the algorithm simpler and faster by dealing directly with the rotor. In this study, to solve the double covering problem involving the rotor, we propose a stateful rotor and develop a technique for uniquely determining the time-varying states of the rotor. The proposed stateful rotor guarantees the continuity of the rotor parameters with respect to angular changes, and this paper confirms its effectiveness by simulating two rotations around an arbitrary axis. In addition, we verify experimentally that a fast sensor fusion method using stateful rotor can be used for attitude calculation. Experiments also confirm that the calculated results converge to the desired rotation angle for two spatial rotations around an arbitrary axis. Since the proposed stateful rotor extends and stabilizes the definition of the rotor, it is applicable to any algorithm that deals with time-varying quaternionic rotors. In this research, an algorithm based on a multiply–add operation is designed to reduce computational complexity as a high-speed calculation for embedded systems. This method is theoretically equivalent to other methods, while contributing to power saving and the cost reduction of products.

## 1. Introduction

Nine-axis sensors, each of which is composed of an accelerometer, magnetometer, and gyroscope, are used to measure the posture and motion of humans and robots [1,2]. Such sensors can either be inertial measurement unit (IMU) sensors or magnetic, angular rate, and gravity (MARG) sensors. Now, to be precise, the IMU is a 6-axis sensor because it does not have a magnetometer, but because the proposed algorithm in this paper is applicable to it, this sensor is regarded without distinction. In any case, the development of micro-electro-mechanical systems has made 9-axis sensors compact and sufficiently popular to be used in smartphones [3] and unmanned aerial vehicles (UAVs) [4,5].

Fast measurement algorithms with small sensors have become an important component of state-of-the-art technology for robot and UAV attitude measurement techniques [6]. In human body motion measurement, and the posture measurement of robots and UAVs, real-time performance is required; thus, a high processing speed is vital. For this reason, we have been developing a fast approximate computation method for nonlinear functions as a fast attitude computation algorithm that can be used in embedded systems [7,8], and have used it in the above-mentioned wearable sensors and powered suits. This has enabled posture measurement on MCUs with small internal memory, where conventional mathematical libraries cannot be incorporated. Using these posture measurement algorithms for embedded systems, we previously developed wearable devices that measure and assist human body motion, mainly in the sagittal plane [9,10,11]. However, although the method of approximating nonlinear functions enabled the fast computation of plane rotations, it was difficult to extend the method to three-dimensional (3D) rotations while keeping computational costs low.

The sensor fusion method algorithm for 3D rotation has been discussed since the 1970s [12]. The goal is to represent the attitude vector in terms of Euler angles, rotation matrices, and pure quaternions, and to eliminate noise from multiple sensors to achieve stable attitude estimation. The implementation methods of this theory include the use of extended Kalman [13,14,15,16,17,18,19], Madgwick [20,21,22,23,24,25] and Mahony filters [26], but our target real-time fast computation for embedded applications requires even lower computational cost and power consumption. Therefore, the design of the proposed algorithm uses a complementary filter that can be processed quickly by embedded computers. Model-based algorithms, such as extended Kalman filters, are difficult to design because the target human body posture measurement and UAV posture are generally difficult to model due to the intervention of the intention of the person being measured and the operator [27]. In contrast, a complementary filter is easy to implement because its frequency characteristics can be controlled via the adjustment of only a single parameter, and simple algorithms can be designed.

Therefore, we propose a sensor fusion algorithm for quaternion rotors, focusing on the fact that the algorithm can be algebraically organized in a simple way by treating quaternions as rotors. If the quaternion rotor is treated as is, sensor fusion cannot be designed due to the problem of guaranteed uniqueness. Therefore, we have developed a stateful rotor that extends the definition of the rotor to solve this problem. The stateful rotor guarantees the continuity of the internal state, allowing the use of statistical and time-series filters. Due to the extension of the definition of rotors, the proposed change to statefulness has the potential to be applied to stabilize various algorithms in addition to the proposed method. In the Results section, we confirm that it is also possible to eliminate time series discontinuities in the rotors computed with conventional algorithms.

The objective of this research is to develop a small, low-consumption, real-time computationally feasible posture measurement sensor that can ultimately be used for human body motion measurements. The proposed method designs the algorithm so that it can be executed only by a multiply–add operation, and run at high speed on MCUs, such as embedded devices. Based on fast computation as a design concept, we used quaternion algebra and complementary filters and designed a stateful rotor and a fast arccos approximation.

## 2. Methods

This section is structured as follows. In Section 2.1, we define the symbols for the quaternions and rotations used in this paper. In Section 2.2, we clarify the problems in designing algorithms for handling rotors and propose a stateful rotor to solve them. In Section 2.3, a fast sensor fusion algorithm using stateful rotor is designed. Finally, in Section 2.4, we propose a linear approximation of a nonlinear function to obtain the attitude angle from a rotor at high speed for embedded systems.

### 2.1. Definition of Quaternion Rotor

The use of quaternions provides advantages such as the absence of singularities and the use of interpolation expressions. These advantages will serve as essential algebraic properties in this algorithmic design. The handling of rotations by quaternions is based on previous studies [28,29,30].

In this paper, the scalar part of a quaternion is represented by *s* or *w*, and the vector part comprises the elements v=(x,y,z). We use the following general definition of a quaternion rotor.
(1)$$\begin{array}{c} q_{r}=\left(s_{r},\;v_{r}\right)=\left(\cos\frac{\theta }{2},\;n\sin\frac{\theta }{2}\right) \end{array}$$

In this definition, θ denotes the angle of rotation, and n denotes the axis of rotation.

The rotation from a point $p=(x_{p},\;y_{p},\;z_{p})$ to another point $p^{\prime }=\left(x_{p}^{\prime },\;y_{p}^{\prime },\;z_{p}^{\prime }\right)$ in 3D space by a quaternion is expressed as
(2)$$\begin{array}{c} q_{p}^{\prime }=q_{r}\;q_{p}\;q_{r}^{*}, \end{array}$$
where the conjugation $q_{r}^{*}$ denotes $\left(s_{r},-v_{r}\right)$ and $q_{p}$ denotes the quaternion; where 0 is added to the scalar part, a pure quaternion is realized, i.e., $q_{p}=\left(0,\;x_{p},\;y_{p},\;z_{p}\right)$. Regarding rotations by quaternions, given that both the rotation operator $q_{r}$ and the point $q_{p}$ of the operation target are four-dimensional quantities, confusion in distinguishing the two concepts can greatly hinder understanding. Therefore, in this paper, the rotor $q_{r}$ and the point $q_{p}$ are regarded separately by using subscripts.

In summary, this section defines the symbols for quaternions and rotors used in this paper. Using these symbols, the proposed algorithmic design is discussed in the next section.

### 2.2. Stateful Rotor

The rotor is defined by Equation (Equation 1). Based on the use of complex numbers in circular statistics, the use of trigonometric functions corresponds to an unwrapping process that smoothly connects −π rad and π rad. Thus, initially, Equation (Equation 1) also seems to guarantee continuity in the quaternion space. However, because the arguments in Equation (Equation 1) are multiplied by a factor of 1/2, the value of θ must range within [−2π,2π] for the trigonometric functions to be continuous over a cycle. In contrast, it is sufficient to consider [−π,π] if we describe every posture only statically. To account for the continuity of the rotor, we need to consider [−2π,2π] because of the influence of the trigonometric function of the definition.

For example, the behavior of each element of $q_{r}$ with respect to the angle n = (0.267,0.535,0.802)∝(1,2,3), when the axis of rotation is θ, is shown in Figure 1. This simulation was calculated when the initial angle was −2π and was varied continuously up to 2π. The dashed lines are the values that must be taken for the rotor to be continuous, whereas the solid lines are the values calculated from Equation (Equation 1). In the range where θ is [−π,π], both the scalar and vector parts appear to be continuous, but the vector part cannot be connected because the signs of the values at π and −π are reversed. This discontinuity has a negative impact on the design of time-series or statistical filters; thus, it is necessary to remove the discontinuity to guarantee continuity.

The problem here is that although θ can represent a 3D rotation in the range [−π,π], if we extend it to [−2π,2π] because of the continuity of the trigonometric functions, there will be two different rotors representing the same attitude. This corresponds to the fact that $q_{r}$ and $-q_{r}$ represent the same attitude, as is evident from the definition of the rotation of quaternions (Equation 2). Whereas uniqueness could be ensured through the selection of either $q_{r}$ or $-q_{r}$ when the domain of definition is [−π,π], the space in which the domain is expanded to [−2π,2π] contains both $q_{r}$ and $-q_{r}$, which leads to arbitrariness with regard to the value that the rotor takes. Figure 1 is a simple case, and thus we can take $q_{r}$ for |θ|≤π and $-q_{r}$ for |θ|>π. In reality, however, θ is an unknown number, and the threshold varies depending on the state of the rotation axis n. Thus, it is difficult to make a decision based solely on this information.

To simultaneously solve the problems of guaranteeing the continuity of trigonometric functions and the uniqueness of the rotor, we propose the use of a stateful rotor. First, the continuity of the trigonometric functions can be guaranteed if the domain of the definition is set to [−2π,2π], as described previously. Next, we consider whether the value computed from the definition of the rotor, expressed by Equation (Equation 1), should be $q_{r}$ or $-q_{r}$. However, because $q_{r}$ and $-q_{r}$ are equivalent informative values, we cannot make a proper choice based on this information alone. Therefore, given that the purpose of this study is time-series signal processing for posture measurement, we consider using dynamic information. Specifically, the optimal state is maintained through the construction of a tracking system that uses the state of one previous rotor to calculate the likelihood. An overview of the designed stateful rotor is shown in Figure 2.

This is ultimately a choice between $q_{r}$ or $-q_{r}$, which can be thought of as a question of whether $q_{r}$ should be multiplied by −1. Therefore, based on a comparison between the currently obtained value $q_{r,i}$ and the past state ${\hat{q}}_{r,i-1}$, it is determined that $q_{r}$ is the optimal value if the distance is close, and $-q_{r}$ is the optimal value if the distance is far. The distance can be evaluated based on the inner product of the elements of $q_{r,i}$ and ${\hat{q}}_{r,i-1}$, that is, the scalar part of the conjugate quaternion product.
(3)J=vec(qr,i)·vec(q^r,i−1)=Re(qr,iq^r,i−1*)=wr,iwr,i−1+xr,ixr,i−1+yr,iyr,i−1+zr,izr,i−1,
where vec is a vectorization that treats elements as vectors. Because this evaluation function is an inner product, it can be computed quickly in the implementation using a multiply–add operation of the elements. Because *J* is an inner product, it has a positive value in the same direction if it is close, and a negative value in the opposite direction if it is far away. Therefore, the sign of *J* can be used to determine the sign of $q_{r}$ as it is.
(4)$$\begin{array}{c} {\hat{q}}_{r,i}=\mathrm{sign}\left(J\right)q_{r,i} \end{array}$$

The ${\hat{q}}_{r,i}$ is a stateful rotor, which guarantees continuity and uniqueness. Examples are shown in Figure 3 and Figure 4. Figure 3 shows the results when the arbitrarily settable initial condition ${\hat{q}}_{r,0}$ and the measurement initial value $q_{r,1}$ are close. Continuity is preserved across the function, and the ends of the domain of definition can also be connected. Figure 4 shows the results when the initial condition ${\hat{q}}_{r,0}$ and ${\hat{q}}_{r,0}$ are far apart. Although the values are inverted, because $q_{r}$ and $-q_{r}$ are equivalent, the inversion of the negative sign is not a problem, and continuity is guaranteed as in the result shown in Figure 3. Therefore, the initial state value can be arbitrary without any problem, and the state-tracking algorithm can obtain a behavior with guaranteed continuity thereafter.

In summary, the discontinuity and uniqueness of the time-varying rotors were identified as problems, which were both solved by introducing the stateful rotor. The stateful rotor is an extension of the definition of rotors; thus, it is an algorithm that can be applied to other research dealing with rotations of quaternions.

### 2.3. Sensor Fusion Algorithm for the Stateful Rotor

Because the stateful rotor guarantees the time-series continuity of the rotor, sensor fusion with time-series filters can be applied. Although previous studies have proposed extended Kalman and Madgwick filters with complementary filters for the sensor fusion of coordinate $q_{p}$, we considered the sensor fusion of $q_{r}$ with the stateful rotor. We considered the application of complementary filters for the purpose of low-cost and fast computation, which was the design concept. The basis of optimization for embedded applications was to eliminate division and nonlinear computation from the algorithm and to use a multiply–add operation to perform the computation. From this perspective, the first-order complementary filter was the best choice because it could be performed using only a multiply–add operation.

Sensor fusion of the rotor with a complementary filter can be designed by analogy with a one-dimensional complementary filter. We designed an algorithm to calculate the rotor via the sensor fusion of 9-axis sensors, using the measured values a of the acceleration sensor, m of the magnetometer, and ω of the gyroscope as inputs.

#### 2.3.1. Low-Frequency Side: Magnetometer and Accelerometer

Gyroscopes measure absolute angular rates in the world frame, and relative value changes in attitude can be calculated from it. In contrast, magnetometers and accelerometers measure absolute attitudes relative to Earth. Methods for obtaining the quaternion attitude include long-established methods, such as the quaternion estimation algorithm (QUEST) and algebraic quaternion algorithm (AQUA); factored quaternion algorithm (FQA) for calculating roll, pitch, and yaw angles; and super-fast attitude from accelerometer and magnetometer (SAAM) for high-speed calculation. Because SAAM has singular postures, as will be shown later, this study used a method based on a direction cosine matrix (DCM), which is as fast as SAAM and as accurate as FQA.

First, we considered normalization and orthogonalization of sensor data. The sensor value matrix as a DCM is as follows.
(5)$$\begin{array}{c} R= [\begin{array}{ccc} m_{x} & c_{x} & a_{x}\\ m_{y} & c_{y} & a_{y}\\ m_{z} & c_{z} & a_{z} \end{array}]= [m\;c\;a]. \end{array}$$

a is the normalized vector of the sensor value of the accelerometer $a_{s}$, and therefore $a=a_{s}/∥a_{s}∥$. c is the virtual axis derived by $c=\left(a\times m_{s}\right)/∥\left(a_{s}\times m_{s}\right)∥$, where $m_{s}$ is the measured value from the magnetometer. Vector c is always orthogonal to a and $m_{s}$; however, a and $m_{s}$ are not guaranteed to be orthogonal. Thus, orthogonality is guaranteed by the cross product of c and a as follows: m=(c×a)/∥(c×a)∥. This technique is known as tri-axial attitude determination [31,32]. This process ensures that *R* is an orthogonal matrix, enabling the transformation formula for rotation matrices and quaternion rotors to be applied. The matrix *R* is referred to as the Earth fixed frame.

Next, the quaternion responsible for the low-frequency side of the complementary filter was generated from the sensor measurements. The method to reconstruct a rotor at high speeds without a singular axis of rotation from DCM is as follows.

First, we consider the element matrix *M* as follows.
(6)$$M= [\begin{array}{cccc} D_{0} & N_{1} & N_{2} & N_{3}\\ N_{1} & D_{1} & P_{3} & P_{2}\\ N_{2} & P_{3} & D_{2} & P_{1}\\ N_{3} & P_{2} & P_{1} & D_{3} \end{array}]= [\begin{array}{cccc} M_{0} & M_{1} & M_{2} & M_{3} \end{array}].$$

This matrix is filled with information on sensor measurements as follows. The diagonal components $D_{i}$ are
(7)$$\begin{array}{c} \begin{array}{cc} D_{0} & =1+m_{x}+c_{y}+a_{z},\\ D_{1} & =1+m_{x}-c_{y}-a_{z},\\ D_{2} & =1-m_{x}+c_{y}-a_{z},\\ D_{3} & =1-m_{x}-c_{y}+a_{z}, \end{array} \end{array}$$
and the nondiagonal components $P_{i}$ and $N_{i}$ are
(8)$$\begin{array}{c} \begin{array}{cc} P_{1}=a_{y}+c_{z}, & N_{1}=a_{y}-c_{z},\\ P_{2}=m_{z}+a_{z}, & N_{2}=m_{z}-a_{x},\\ P_{3}=c_{x}+m_{y}, & N_{3}=c_{x}-m_{y}. \end{array} \end{array}$$

Next, we reconstructed the quaternion rotor from *M*. We only used a one-column vector $M_{k}$. The *k* is the index that maximizes $D_{i}$, i.e., $k={arg max}_{i}\left(D_{i}\right)$. The quaternion rotor $q_{rs}$ was reconstructed from the component vector $v_{qs}$ as follows:(9)$$\begin{array}{c} v_{qs}={\left.\frac{1}{2\sqrt{D_{k}}}M_{k}\right|}_{k=\mathrm{argmax}\left(D_{i}\right)}. \end{array}$$

The quaternion rotor $q_{rs}$ reconstructed from the component vector is
(10)$$\begin{array}{c} q_{rs}=v_{q}\cdot v_{qe}. \end{array}$$

Here, $v_{qe}$ is a vector that has the unit elements of the quaternion, i.e., $v_{qe}=\left[1,\;i,\;j,\;k\right]$.

With regard to the programming, all $D_{i}$ were first calculated, and *k* was determined as $k=\mathrm{argmax}(D_{i})$. The *k*-th column vector of the element matrix *M* was then calculated and divided by $2\sqrt{D_{k}}$. Then, we could obtain the four components of the desired quaternion. Its calculation cost was 13 additions, 5 multiplications, 1 conditional branch, and 1 reciprocal square root. For example, if the fast inverse square root algorithm [33,34,35,36,37,38] was used for the reciprocal square root, this algorithm could be executed using only multiply–add operations.

#### 2.3.2. High-Frequency Side: Gyro Sensor

Next, we consider how to treat the gyro sensor value ω as a rotor.

If the rotation ω measured by the gyro sensor is the rotor $q_{r\omega }$, then based on the definition of the gyro sensor measurement, the rotation angle per minute time is ∥ω∥Δt, and the rotation axis is $\frac{\omega }{∥\omega ∥}$. Therefore, $q_{\omega }$ could be obtained from the definition of the rotor Equation (Equation 1) as
(11)$$\begin{array}{cc} q_{r\omega } & =\left(\cos\frac{∥\omega ∥\Delta t}{2},\;\frac{\omega_{x}}{∥\omega ∥}\sin\frac{∥\omega ∥\Delta t}{2},\frac{\omega_{y}}{∥\omega ∥}\sin\frac{∥\omega ∥\Delta t}{2},\;\frac{\omega_{z}}{∥\omega ∥}\sin\frac{∥\omega ∥\Delta t}{2}\right). \end{array}$$

Using the small-angle approximation sinθ≈θ(θ≪1) and the rotor condition $∥q_{r}∥=1$, we obtain
(12)$$\begin{array}{cc} q_{r\omega } & =(s_{\omega },v_{\omega })\\ \; & \approx \left(1-\frac{∥v_{\omega }{∥}^{2}}{2},\;\frac{\omega_{x}\Delta t}{2},\frac{\omega_{y}\Delta t}{2},\;\frac{\omega_{z}\Delta t}{2}\right). \end{array}$$

Therefore, the integration of the attitude by the gyroscope became the following update rule.
(13)$$\begin{array}{cc} q_{p,n} & =\left(q_{r\omega ,n}\right)\;q_{p,n-1}\;\left(q_{r\omega ,n}^{*}\right)\\ \; & =\left(q_{r\omega ,n}\;q_{r\omega ,n-1}\right)\;q_{p,n-2}\;\left(q_{r\omega ,n-1}^{*}\;q_{r\omega ,n}^{*}\right)\\ \; & =\left(\prod_{i=1}^{n}q_{r\omega ,i}\right)\;q_{p,0}\;{\left(\prod_{i=1}^{n}q_{r\omega ,i}\right)}^{*} \end{array}$$

From this, the update rule for the rotor by the gyro sensor is
(14)$$\begin{array}{c} q_{r,n}=q_{r\omega ,n}\;q_{r,n-1}. \end{array}$$

The expression could have a simplified form because it needs only Equations (Equation 12) and (Equation 14).

#### 2.3.3. Sensor Fusion with Complementary Filter

Finally, $q_{rs}$ and $q_{r\omega }$ are combined by a complementary filter. Because of the characteristics of the sensor, the high-frequency side of $q_{rs}$ is subject to motion acceleration noise among others, whereas the low-frequency side is highly reliable. On the other hand, the low-frequency side of $q_{r\omega }$ accumulates integration errors, whereas the high-frequency side is highly reliable. The complementary filter is a technique that uses only the frequency responses of these reliable portions of the data and filters the bands with the most errors [27,39,40,41]. The complementary filter can be calibrated using α, a cut-off frequency control parameter that takes a value from 0 to 1, as follows.
(15)$$\begin{array}{c} q_{rc,i}=\alpha q_{r\omega }\;q_{rc,i-1}+\left(1-\alpha \right)q_{rs} \end{array}$$

The first term on the right-hand side is an updated expression based on the integration of the gyro rotation, whereas the second term on the right-hand side represents the current attitude estimate obtained from the Earth fixed frame. Through interpolation by α, the frequency characteristics were improved, and the error was converged to zero.

Here, $q_{rc,i}$ and $q_{rs}$ appear on the right-hand side. Therefore, as discussed on the stateful rotor in Section 2.2, because $q_{rs}$ has the degrees of freedom, $q_{rs}$ and $-q_{rs}$, the entire system becomes unstable if the one far from $q_{rc,i}$ is selected. Therefore, stabilization is performed via the transformation of $q_{rs}$ to statefulness, with $q_{rc,i}$ as the state, as shown in Figure 5.

Finally, sensor fusion could be performed such that the error converges to zero as in the following equation, resulting in a time-series stable rotor.
(16)$$\begin{array}{c} {\hat{q}}_{rc,i}=\alpha q_{r\omega }\;{\hat{q}}_{rc,i-1}+\left(1-\alpha \right){\hat{q}}_{rs} \end{array}$$

To summarize this subsection, a sensor fusion method for the 9-axis sensors using a stateful rotor is proposed. The algorithm developed adheres to the design concept of implementation by a multiply–add operation as much as possible, and the calculation method is suitable for embedded systems.

### 2.4. Fast Angle Estimation from Rotor

We propose a method to linearly approximate the nonlinear operations required in the angle calculation with high accuracy in order to speed up the algorithm for embedded systems. This removes the nonlinear computation from the entire attitude estimation algorithm and optimizes it for embedded systems with limited computational resources.

In the previous section, to update the rotation state at high speeds, sensor fusion of the rotor using quaternions was performed. Ultimately, the rotation state could be intuitively determined if the rotation angle and axis of rotation were obtained from the rotor.

Based on the definition of the rotor Equation (Equation 1), the angle of rotation and the axis of rotation are
(17)$$\begin{array}{cc} \theta & =2\arccos(w), \end{array}$$
(18)$$\begin{array}{cc} n & =\frac{1}{\sin\frac{\theta }{2}}{ [x,\;y,\;z]}^{T}=\frac{1}{\sqrt{1-w^{2}}}{ [x,\;y,\;z]}^{T}. \end{array}$$

Although n could be calculated using only elements and inverse square roots, the arccos operation, a nonlinear function, is required to calculate θ. Given the use of low-performance MCUs for downsizing wearable sensors and UAVs, the use of nonlinear functions should be avoided as much as possible. Therefore, this nonlinear function is processed by approximation to achieve high speeds. The residual correction method proposed in our previous study [7,8] allowed calculations using only square root operations, as shown in the following equation. When β=0.351, the maximum approximation error was less than 0.5 deg as showin in Figure 6.
(19)$$\begin{array}{cc} \theta & =2\arccos\left(w\right)\approx \left\{\begin{array}{cc} \left(\pi -\beta w\right)\sqrt{1-w} & \mathrm{if}\;w\geq 0\\ 2\pi -\left(\pi +\beta w\right)\sqrt{1+w} & \mathrm{otherwise} \end{array}\right. \end{array}$$

In summary, we use the fast inverse square root algorithm to represent everything from sensor measurements to attitude computation in a multiply–add operation. This enables fast and low-power computation, even on embedded systems with limited computational resources.

## 3. Results

We confirm the effectiveness of the proposed stateful rotor and fast sensor fusion by using it. In Section 3.1, we confirmed through simulation that the stateful rotor guaranteed continuity of all four parameters of the rotor with respect to the time variation of the angle. In Section 3.2, we confirmed that sensor fusion using stateful rotor worked correctly for arbitrary 3D rotations through experiments using actual sensors.

### 3.1. Effects of Stateful Rotor

We performed simulations to compare the quaternion $q_{rs}$ reconstructed from the accelerometer and magnetometer before and after transformation to statefulness. The item to be verified here was the four parameters of the quaternion by using stateful rotor maintained continuity for time-varying angles. In addition to the DCM used for sensor fusion, we also applied the stateful rotor to FQA and SAAM to confirm its applicability to other methods.

The transformation to statefulness was confirmed to remove rotor discontinuity for when the angle θ changes continuously. The angle was varied continuously from −360 deg to 360 deg, and the axis of rotation was n∝(1,2,3) from −360 deg to −120 deg and n=(0,0,1) from 120 deg to 360 deg. It also varied continuously from −120 deg to 120 deg in between. The initial value of state was ${\hat{q}}_{r,0}=\left(1,\;0,\;0,\;,0\right)$.

The respective stateless and stateful results are shown in Figure 7, Figure 8 and Figure 9.

According to the results, DCM and FQA have the same calculation results; whereas statelessness causes a discontinuity point, a transformation to statefulness makes the change continuous. By comparison, SAAM is faster but has a singular rotational axis, and the calculation results at approximately n=(0,0,1) become $q_{r}=\left(0,\;0,\;0,\;0\right)$. Therefore, by calculating the rotor based on a change to statefulness, the method based on DCM without a singular rotation axis is suitable for embedded applications with a minute amount of computation, which is approximately the same as that for SAAM.

### 3.2. Sensor Fusion of Rotor with Complementary Filter

Next, we experiment with the fast computation of sensor fusion and angle estimates for the rotor using complementary filters based on conversion to statefulness. A DCM-based method is used to obtain $q_{rs}$ from the accelerometer and magnetometer. The verification item from this experiment is to confirm that the four parameters of the respective rotors of $q_{rs}$ and $q_{r\omega }$ are restored to the desired values. We also confirm that the angle measurements are correct when they are combined by sensor fusion.

Because the dynamic characteristics of the complementary filter have to be verified, experiments were conducted using actual equipment. The sensor was an MPU9250 (InvenSense) with a sampling time of dt=0.1 s. The device used in the experiment is shown in Figure 10 and the measured values from the sensor are shown in Figure 11.

The sensors were calibrated using only the values on the datasheet, and were not calibrated for individual differences. The first half of the experiment involved a rotation around the y-axis, whereas the second half involved a rotation around the z-axis, each ranging 360 deg. In this experiment, the sensor was held in the hand at a speed of one rotation approximately every four seconds. Therefore, the frequency response α of the complementary filter was experimentally searched to be able to follow the frequency of such human motion.

Based on these data, the results of the stateless complementary filter are shown on the left side of Figure 12. The stateless complementary filter synthesized the measured values with different frequency characteristics by a first-order infinite impulse response low-pass filter and high-pass filter to keep the gain at one. Therefore, as a design principle, the two values to be synthesized should have the same trend. However, according to the left side of Figure 12, the values of $q_{rs}$ and $q_{r\omega }$ are completely different, and thus the synthesis failed.

Next, the results of the computation of the rotor by the complementary filter when it was converted to statefulness are shown in the right side of Figure 12. The continuity of $q_{rs}$ is guaranteed, eliminating discontinuities, and the behavior is close to that of $q_{r\omega }$ calculated by fast quaternion integration based on the gyro sensor measurements. Therefore, these could be synthesized simply by a complementary filter using the regions where each frequency characteristic is strong, and the rotor update calculation could be performed reliably.

## 4. Discussion

In this section, we first discuss, from the experimental results, that sensor fusion at the rotor stage, which could not be performed by the conventional method (stateless), is now possible by using a stateful rotor. Next, we discuss the objectives of this study, namely, the reduction in computational cost and implementation of a multiply–add operation.

### 4.1. Usefulness of Conversion to Statefulness

Through the conversion of the rotor to statefulness, ${\hat{q}}_{rs}$ was calculated quickly without a singular rotation axis using DCM, whereas ${\hat{q}}_{rc}$ was calculated via a combination of gyro sensor integration with fast quaternion integrating with a complementary filter, and its effect was confirmed through experiments. Herein, the conversion from rotor to attitude angle estimates, which was the objective, is considered. For sagittal surfaces, arctangent calculations can be used after projection, but in this case, a fast arccos approximation method, expressed as Equation (Equation 19), is used. Figure 13 shows the calculation results.

As shown in the results, for the stateless rotor and because of the state ambiguity caused by the double covering problem, the composite by the complementary filter failed. This failure resulted in angle estimates that were far from the true value or the gyro-integral value, which was close to the true value. Therefore, the conventional method stateless rotor can only be used to the extent that it does not exceed the point of discontinuity. On the other hand, the stateful rotor is able to track the rotation state well, even after a 360-deg rotation, and the estimated attitude angle is close to the gyro-integral value. In addition, the effect of the complementary filter eliminates the integration error remaining in the gyro-integral value, indicating that the sensor attitude angle can be estimated with good frequency response.

Next, a more specific example of measured thigh motion during walking is shown in Figure 14. This walking motion is measured simultaneously with an optical motion capture system. The results show that IMU using the proposed stateful algorithm is able to measure the same motions as motion capture. However, the stateless algorithm shows that there is an attitude that makes the calculation wrong when it turns around. This is due to the jump in states shown in the simulation, and we have confirmed that the stateful algorithm can solve this problem.

In summary, we confirmed that sensor fusion failed with the conventional using the stateless rotor, but stable attitude measurement was possible with the proposed stateful rotor. For limitation, there is no restriction on the rotational motion because the algorithm measures two rotational movements in arbitrary axes this time, and there are no singularities or gimbal locks, and the algorithm supports infinite rotation. In addition, for human body measurement applications, wearable IMUs do not have the limitations of the measurement area as optical motion capture has. In this time, experimental parameter search was conducted so that the frequency response of the complementary filter would be optimized by human motion. Although it cannot be adaptively optimized like the Kalman gain of a Kalman filter, it has a degree of freedom as a design parameter in measurements where the frequency response can be estimated.

### 4.2. Calculation Cost

Herein, the computation speed is considered. In an embedded system, the computation speed is proportional to the amount of central processing unit (CPU) time required for an operation, and because the CPU time varies depending on the capabilities of the CPU and microcontroller unit (MCU) used, we count the number of operations required for a computation.

First, the computational complexity of finding $q_{rs}$ from magnetometer and acceleration sensor measurements is determined. This is then compared with the computational complexity of the algorithm after orthogonalization and normalization of the sensor data. The results are shown in Table 1. The proposed algorithm is as fast as SAAM, and by using the fast inverse square root, all calculations can be performed using multiply–add operations.

Next, we consider the computational cost of the gyro-integral and complementary filters required for sensor fusion. With regard to updating based on $q_{p}$, there were many methods that treated ω as an antisymmetric matrix [3,43,44,45,46,47,48,49,50]. Almost all the methods, from those designed by Bortz in the 1970s [12] to those in the 2020s, used this technique. However, this approach is computationally expensive. In contrast, if the rotor $q_{r}$ is used as a base, the “Fast Quaternion Integration for Attitude Estimation” approach [51,52], which is an intuitive update law based on the definition of the rotor, becomes available.

The computational cost of the gyro-integral and complementary filters required for sensor fusion is shown in Table 2.

Conversion to statefulness is also a multiply–add operation because it is the inner product of vectors, and thus the entire algorithm can be expressed as a multiply–add operation. The conversion to angles can also be calculated via the residual correction method using Equation (Equation 19). Therefore, even MCUs that do not have floating-point units can perform attitude calculations at high speeds.

In summary, the computational cost of the proposed method was discussed. The computation cost of $q_{rs}$, which is the most computationally expensive, was comparable to that of other optimal methods and can be implemented entirely by a multiply-add operation, which is fast. In addition, since a complementary filter is used for sensor fusion, this can also be performed using only a multiply–add operation, making it an optimized structure for embedded systems. The final angle calculation can also be performed only by a multiply–add operation using our fast approximation of nonlinear functions. Therefore, by optimizing all calculations using a multiply–add operation, it is an optimal attitude calculation algorithm for embedded systems.

## 5. Conclusions

In this study, for the purpose of high-speed computation for embedded systems using low-performance CPUs, such as MCUs, $q_{rs}$ was computed quickly from sensor measurements using DCM, and gyro-sensor values were processed using fast quaternion integration for complementary filtering. Using the fast inverse square root algorithm and other algorithms, all of these could be easily computed using only multiply–add operations, making the algorithm suitable for embedded systems.

As shown in one example in the experiment, the proposed method has the potential to be used for stable and fast measurement of human body motion. In particular, because the proposed algorithm does not have a singular posture, it is considered effective for complex three-dimensional movements in which the axis is not fixed, such as upper body rotational movements. For future work, we will develop a wearable sensor as a small embedded system to measure more complex human body movements and investigate the range of applicability and limitations of the proposed method.

## Figures and Tables

**Figure 1 sensors-22-07989-f001:**
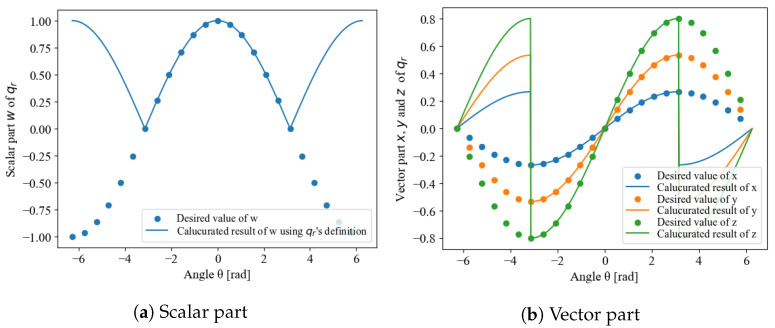
Incontinuity of rotor’s definition.

**Figure 2 sensors-22-07989-f002:**
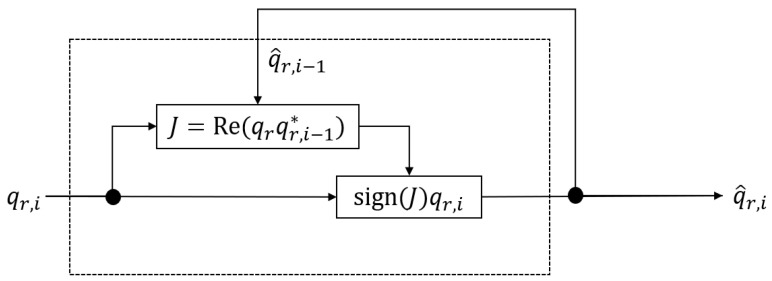
Block diagram of stateful rotor.

**Figure 3 sensors-22-07989-f003:**
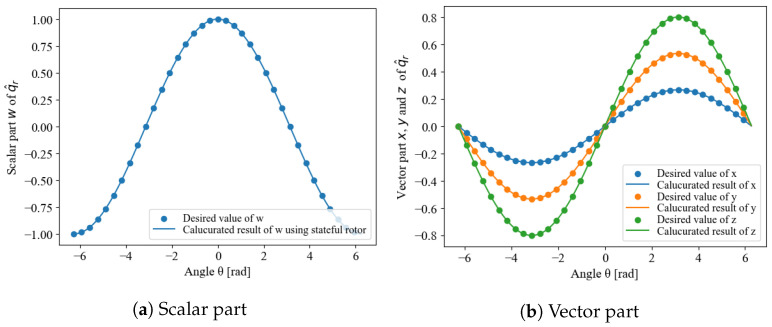
Continuity and uniqueness of stateful rotor when the initial state ${\hat{q}}_{r,0}$ is close to $q_{r,1}$.

**Figure 4 sensors-22-07989-f004:**
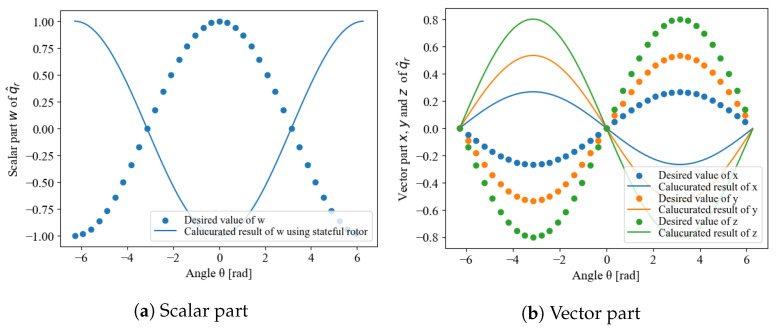
Continuity and uniqueness of stateful rotor when the initial state ${\hat{q}}_{r,0}$ is far from $q_{r,1}$.

**Figure 5 sensors-22-07989-f005:**
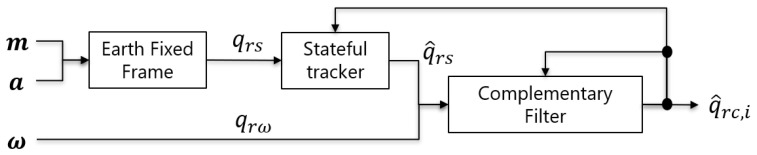
Block diagram of stateful complementary filter.

**Figure 6 sensors-22-07989-f006:**
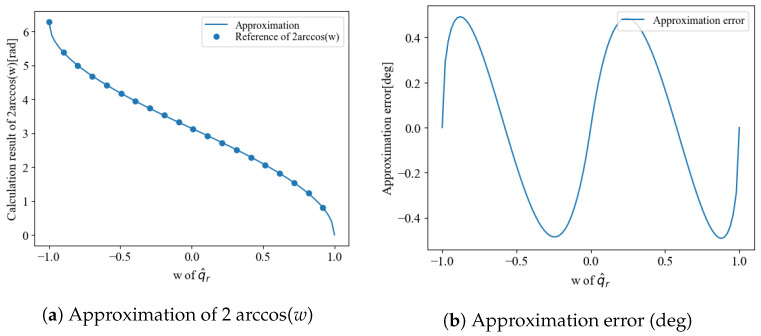
Approximation for fast posture estimasion.

**Figure 7 sensors-22-07989-f007:**
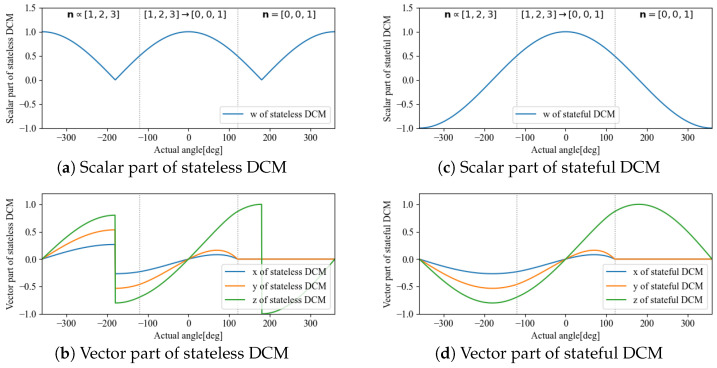
Comparison between stateless DCM (**left**) and stateful DCM (**right**).

**Figure 8 sensors-22-07989-f008:**
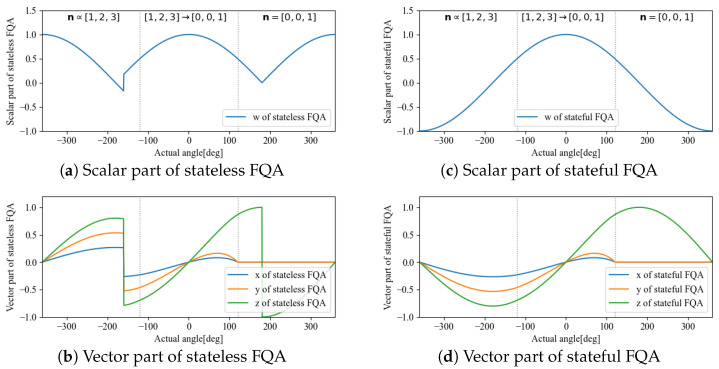
Comparison between stateless FQA (**left**) and stateful FQA (**right**).

**Figure 9 sensors-22-07989-f009:**
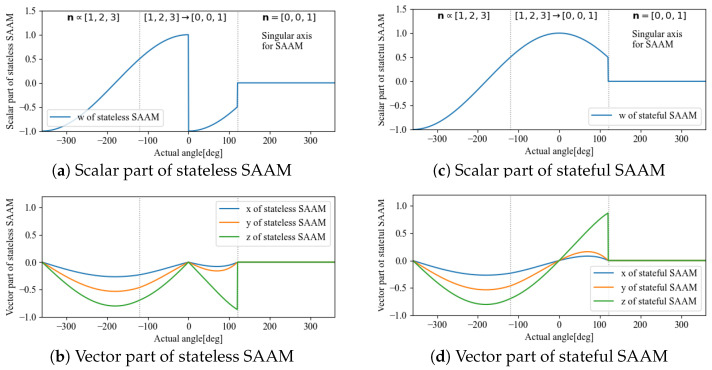
Comparison between stateless SAAM (**left**) and stateful SAAM (**right**).

**Figure 10 sensors-22-07989-f010:**
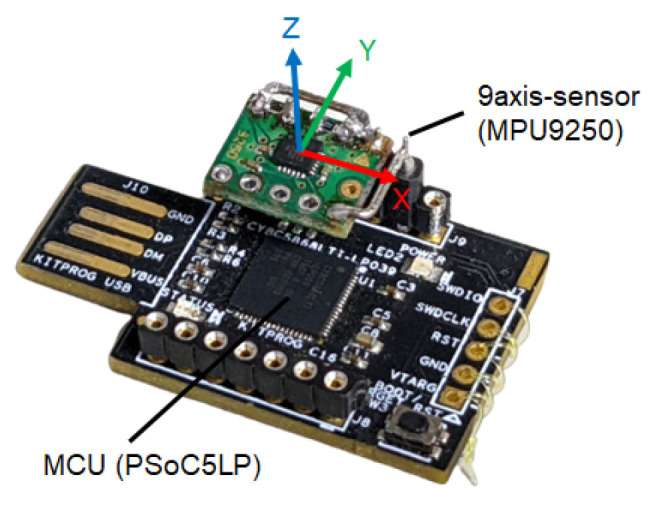
Experimental device for retrieving 9-axis sensor data.

**Figure 11 sensors-22-07989-f011:**
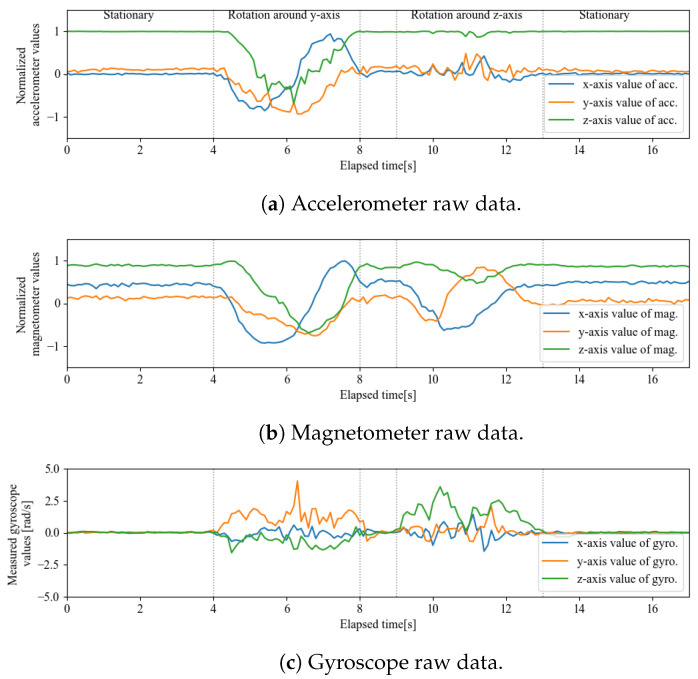
Experimental data obtained from actual equipment (MPU9250).

**Figure 12 sensors-22-07989-f012:**
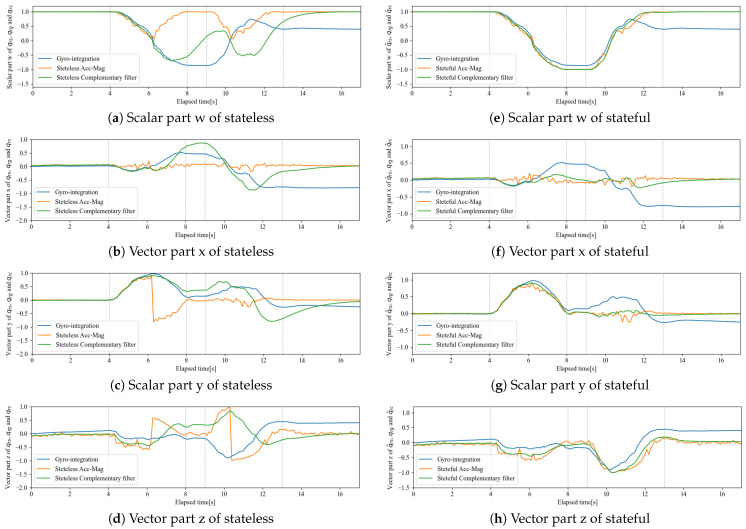
Comparison of sensor fusion results between stateless and stateful rotors.

**Figure 13 sensors-22-07989-f013:**
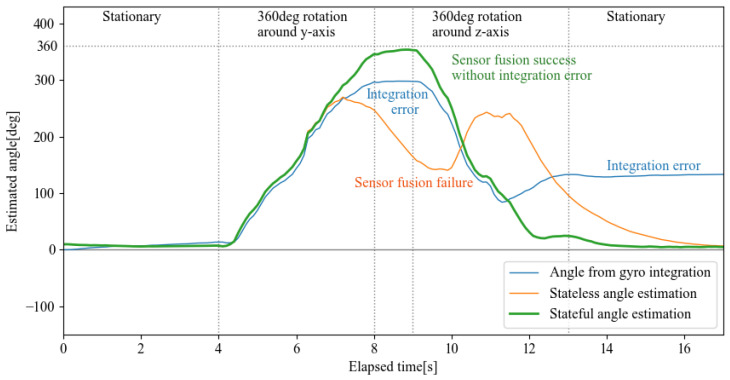
Fast angle estimation for stateful rotor.

**Figure 14 sensors-22-07989-f014:**
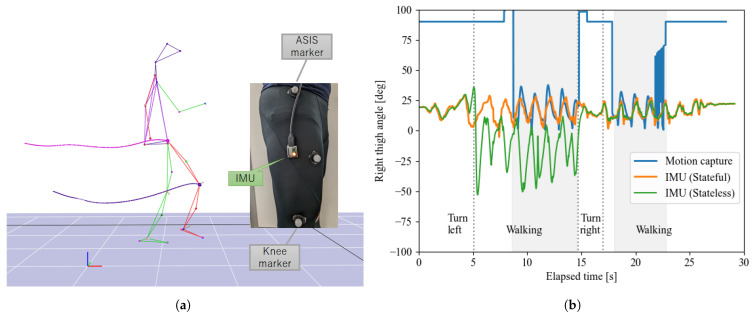
Measurement of walking motion using wearable IMU and optical motion capture. (**a**) Motion capture experiment (IMU placed on right thigh). (**b**) Comparison among motion capture and stateful/less IMU (shaded areas are within the motion capture measurement volume).

**Table 1 sensors-22-07989-t001:** Comparison of calculation cost of $q_{rs}$ (results of SAAM and FQA are cited from [42]).

	+	×	/	√	cos·
DCM-based without fast inverse square root	13	5	1	1	0
DCM-based with fast inverse square root	15	9	0	0	0
SAAM	18	16	1	2	0
FQA	18	53	4	3	6

**Table 2 sensors-22-07989-t002:** Calculation cost of complementary filtering.

	+	×	/	$\sqrt{\cdot }$
Fast quaternion integration	3	7	0	0
Complementary filter	24	16	0	0

## Data Availability

Not applicable.

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
