# Peer review of "Stateful Rotor for Continuity of Quaternion and Fast Sensor Fusion Algorithm Using 9-Axis Sensors"

_sensors, 2022, doi:10.3390/s22207989_

Round 1
Reviewer 1 Report
The paper discuss a new approach for handling 9-axis sensor data. My major concern is the structure of the paper, which needs to be improved.
Abstract: the abstract: ”through simulations and experiments” – Please describe which simulations and experiments that were done to analyse your hypothesis/research question.
General comment: You introduce many abbreviations: IMU, MARG, UAV, ROS, SOQUEST, FQA, IIR-HPF, IIR-LPF, MCU, SAAM, FISR…. Are all abbreviations really needed? Too many abbreviations hinders readers from understanding your study.
1. Introduction:
- You have a very brief introduction on other studies in this area with a few references (line 23-31) followed by a description of your own work (line 32-35). Please extend the introduction on the research done in this area and why/how current algorithms are not successful and a new algorithm is needed.
- Clarify the aim of this specific study. You propose a new method, but why and what do you expect the method to do which is better that earlier methods?
2. Three-dimensional rotation by quaternion:
- is this section really needed? The description on quaternions could instead be referred to (e.g a book or study). In any case, the first sentences (line 55-60) can be removed. Also, you introduce the equations without explaining all components, for example w and v. I suggest you to reduce this section or move the most important parts to the Method section.
3. Related studies and proposed method
- 3.1 other methods should be removed and instead merged into the introduction.
- 3.2 proposed method contains the hypothesis which should have been presented together with the aim in the introduction.
- The method section is long and discuss different approaches and simulations. Please construct the method so that it is more clear to the reader which simulations and experiments you have done in order to test you hypothesis in order to answer you aim.
- Which kind of movements have you analysed? It is not clear which parts of the results that are simulations and which parts are experimental data. Which kind of data are the algorithms suitable for, certain human movements (Gait? Running? Fast/slow movements?) or any movements in general?
4. Discussion
- You should discuss whether the aim was reached, and also compare your method’s strength’s and weaknesses in comparison to other studies. I do not find any references to own or other people’s work in the short discussion section.
Reviewer 2 Report
1. It seems the submission concerns the attitude as a rotation vector. Bortz wrote about this in 1971. The authors should mention his work in the Introduction.
2. Authors wrote that 'gyroscopes measure relative value changes in attitude' (p.6). Gyroscopes measure absolute angular rate.
3. The Results (item 6) should be expanded. The comparison of scalar and vector parts is not enough.
4. Authors used a complementary filter. What does it mean? It should be cleared.
5. A formula (1) has been presented without any reference. Perhaps the second line of this expression is not correct.
Round 2
Reviewer 1 Report
The paper has improved, and the authors have met most of the considerations from the first review. Some issues remain. Please see the comments below.
Abstract:
· The abstract is too long, 200 words maximum.Please consider the journal’s instructions, especially “Place the question addressed in a broad context and highlight the purpose of the study”. (“We needed a technology for real-time measurement of 3D human body motion.” is not a broad context)).
Introduction:
· · You include many self-citations (reference 7-13). I advise you to remove the sentence on row 39-40 and instead focus on the following two sentences including your reference 12-13 that relate directly to this specific paper. Please consider the journal’s instructions for the introduction: “The current state of the research field should be reviewed carefully and key publications cited.”
Method
· - The method section has improved, but include sections that perhaps belong better in the introduction (for example line 161-166) and part that is better placed under discussion (for example line 201-207).
Discussion:
· You do not discuss your results and your methods’ strengths and limitations in relations to previous studies or similar algorithms by other researchers. I cannot find any citations at all. I therefor do not think the discussion meet the journals instruction on the Discussion section: “Authors should discuss the results and how they can be interpreted in perspective of previous studies and of the working hypotheses. The findings and their implications should be discussed in the broadest context possible, and limitations of the work highlighted.”
· - You analyse simulated data. How generalizable are your results? How would the computations work on data from e.g. a gait or running task?
Conclusion:
· Consider reducing the conclusion, since you repeat a lot of the discussion/introduction here. I suggest to remove row 355-360, and row 367-370 , and only keep row 361-366 + the last sentence, and you could complete with “, but the algorithm first need to be evaluated using real human motion data”
Reviewer 2 Report
It is recommended to accept the publication.
Author Response
Dear Reviewer 2,
Thank you very much for reviewing this paper.
Another reviewer has commented again and we have improved the manuscript by adding new experiment.